# The Effect of Y Doping on Monoclinic, Orthorhombic, and Cubic Polymorphs of HfO_2_: A First Principles Study

**DOI:** 10.3390/nano12234324

**Published:** 2022-12-05

**Authors:** Eleonora Pavoni, Elaheh Mohebbi, Davide Mencarelli, Pierluigi Stipa, Emiliano Laudadio, Luca Pierantoni

**Affiliations:** 1Department of Materials, Environmental Sciences and Urban Planning, Marche Polytechnic University, 60131 Ancona, Italy; 2Information Engineering Department, Marche Polytechnic University, 60131 Ancona, Italy

**Keywords:** HfO_2_, Yttrium doping, DFT, Ab-initio, polymorphs

## Abstract

HfO_2_ can assume different crystalline structures, such as monoclinic, orthorhombic, and cubic polymorphs, each one characterized by unical properties. The peculiarities of this material are also strongly related to the presence of doping elements in the unit cell. Thus, the present paper has the main purpose of studying and comparing twelve different systems characterized by diverse polymorphs and doping percentages. In particular, three different crystalline structures were considered: the monoclinic *P2_1_/c*, the orthorhombic *Pca2_1_*, and the cubic *Fm$\bar{3}$m* phases of HfO_2_. Each one has been studied by using Y as a doping agent with three different contents: 0% Y:HfO_2_, 8% Y:HfO_2_, 12% Y:HfO_2_, and 16% Y:HfO_2_. For all the systems, density functional theory (DFT) methods based on PBE/GGA, and on the HSE hybrid functionals were used to optimize the geometry as well as to study their optical properties. Depending on the polymorphs, Y affects the formation energy in different ways and causes changes in the optical properties. When the percentage of Y did not exceed 12%, a stabilization of the cubic phase fraction and an increase of the dielectric constant was observed. Additionally, the calculated optical bandgap energies and the refractive index are examined to provide an overview of the systems and are compared with experimental data. The bandgaps obtained are in perfect agreement with the experimental values and show a slight increase as the doping percentage grows, while only minor differences are found between the three polymorphs in terms of both refractive index and optical band gap. The adopted first principles study generates a reasonable prediction of the physical-chemical properties of all the systems, thus identifying the effects of doping phenomena.

## 1. Introduction

Hafnium oxide (HfO_2_) is an inorganic compound widely used and applied in the semiconductor industry thanks to its numerous peculiarities. HfO_2_ is characterized by a large bandgap and a tuneable dielectric constant that are important for creating an alternative to SiO_2_. Moreover, HfO_2_ allows us to overcome some issues related to perovskites-based field-effects transistor (FET) technologies, it displays a full complementary metal-oxide-semiconductor (CMOS) compatibility, and it has already been introduced as high-k material in a manufacturing process by Intel in 2007 [1]. As already mentioned, HfO_2_ has a relatively wide bandgap, a large band offset with Si (less parasitic leakage), and low permittivity; the absence of an interfacial dead layer in HfO_2_ makes this material a promising candidate in thin-film technology, unlike perovskites-based materials [2,3,4]. Most of the applications related to HfO_2_ are based on the ferroelectric nature of its polymorphs; for example, hafnia is used in ferroelectric random-access memory (FeRAM) and ferroelectric field-effect transistor (FeFET) [5,6]. However, HfO_2_ can also be suitable for infrared (IR) sensors, pyroelectric energy harvesters, and solid-state cooling devices [7,8,9,10].

HfO_2_ is present in nature as different polymorphs, including the monoclinic *P21/c* phase (stable at room temperature), the tetragonal *P42/nmc* and cubic *Fm*$\bar{3}$*m* phases (both stable at high temperature), the orthorhombic *Pbca* (stable at high-pressure), and the disordered *Pbcm* and *Pmnb* phases; among them, the monoclinic one (space group *P2_1_/c*) is the most stable at ambient conditions, but for other crystalline structures, for example, the non-equilibrium polar orthorhombic *Pca2_1_* phase is very important because it is responsible for the observed ferroelectricity in hafnia [11,12]. The crystalline phases could be stabilized by different factors. It has been already shown that HfO_2_ undergoes a transition from monoclinic to cubic at high temperatures [13,14] and pressures. Indeed, in different studies, the high pressure Raman spectra at room temperature on HfO_2_ single-crystal were reported up to 50 GPa [15], followed by three pressure-induced phase transitions at 4.3 GPa, 12 GPa, and 28 GPa. It is also known that the first two transitions are reversible, whereas the third one is irreversible [16]. Another important concern is the presence of doping elements (such as Zr atoms) that could bring a stabilization of the orthorhombic polymorph [17].

The conversion from the monoclinic to other polymorphs, such as the already cited cubic (*Fm*$\bar{3}$*m*) or orthorhombic (*Pca2_1_* non-centrosymmetric space groups), is of primary importance due to the change in the dielectric constant values; in particular, the dielectric constant increases when moving from the monoclinic to the orthorhombic or cubic phases, making the material even more attractive for nanotechnological applications.

To avoid the use of a high-temperature synthetic procedure, different doping elements in different concentrations have already been considered [17,18,19].

The presence of Yttrium as a dopant in HfO_2_ and the effect on the optical and electronic properties, as opposed to what is already known for other elements, is less investigated. As explained by Rauwel et al., the addition of Y brings a stabilization of the cubic phase of HfO_2_ in film grown by chemical vapor deposition [20]. Chen et al. studied the role of Y_2_O_3_ in the microstructure and in the crystallinity degree of HfO_2_ thin film; they also investigated the optical properties at different concentrations of Y_2_O_3_ dopant [21]. Similarly, Liang et al. [22,23] investigated the properties of Y-doped HfO_2_ using various Y content and film thicknesses. Other authors considered the effect of Y-doped HfO_2_ and its relation with the ferroelectricity behavior;, they studied the hysteresis loop, the coercive field, and the polarization of HfO_2_ thin film doped with different amounts of Y [11,24,25,26]. Padilha and McKenna [27] used first-principles calculations to elucidate Y_2_O_3_ doping HfO_2_ of monoclinic and cubic structures.

In the present work, we report a wide theoretical study regarding the three polymorphs of HfO_2_ and the effect of different concentrations of Y as a doping element. Thus, twelve different systems are presented: the monoclinic (m-) with a space group *P2_1_/c*, the orthorhombic (o-) with a space group *Pca2_1_*, and the cubic phase (c-) with a space group *Fm*$\bar{3}$*m* crystalline structures of (i) pure HfO_2_ and Y-doped HfO_2_ after the addition of (ii) 8% of Y, (iii) 12% of Y, and (iv) 16% of Y.

A wide comparison of different properties has been performed using the Density Functional Theory (DFT) approach based on generalized gradient approximation (GGA) exchange-correlation functional [28]. Even if the DFT approach is known to be reliable for ground-state properties predictions, different studies indicate that conventional exchange-correlation (xc) functionals often underestimate the bandgap of semiconductors in experimental studies [29]. To overcome this problem, the HSE hybrid functional has been used to predict the optical band gap energy for all the reported systems.

## 2. Methods

All simulations were performed by using Quantum Atomistic Toolkit (Q-ATK) [30] and Quantum ESPRESSO (QE) [31] atomic-scale modelling. All the twelve structures based on monoclinic (*P2_1_/c*), orthorhombic (*Pca2_1_*), and cubic (*Fm*$\bar{3}$*m*) polymorphs were modelled considering: (i) HfO_2_ with 0% of Y substitution, (ii) HfO_2_ with 8% of Y substitution, (iii) HfO_2_ with 12% of Y substitution, and (iii) HfO_2_ with 16% of Y substitution with respect to the total Hf amount (the Hf ions were randomly substituted with Y in all twelve models). The single particle wave-functions were expanded on the basis of the plane-wave (PW) method for all the Hf, Y, and O entities not dissimilar to the SIESTA formalism [32] For the electron xc energy, the calculations were carried out using Perdew–Burke–Ernzerhof (PBE) GGA density functional [33] For each element, the ionic cores were represented by norm-conserving (NC) PseudoDojo (PDj) pseudopotentials [34] Then regarding the valence electrons, the 5d^2^ and 6s^2^ electrons of Hf and 4d^1^ and 5s^2^ electrons for Y were explicitly treated as valence; obliviously, the 2s^2^ and 2p^4^ electrons were taken as the valence electrons for O atoms. To estimate the imaginary part of the dielectric constant the Heyd–Scuseria–Ernzerhof (HSE) [35,36] hybrid functional has been used; this latter uses an error-function-screened Coulomb potential, has the following form (1):(1)$$E_{XC}^{HSE}=\alpha E_{X}^{HF,SR}\left(\omega \right)+\left(1-\alpha \right)E_{X}^{PBE,SR}\left(\omega \right)+E_{X}^{PBE,LR}\left(\omega \right)+E_{C}^{PBE}$$
where *α* is a mixing parameter and *ω* is an adjustable parameter controlling the short range of the interaction. Standard values of *α* = 1/4 and *ω* = 0.2 usually have been shown to give good results for most of the systems. In this equation, $E_{X}^{HF,SR}\left(\omega \right)$ and $E_{X}^{PBE,SR}\left(\omega \right)$ are referred to short-range Harteree–Fock (HF) and PBE exchange functional, $E_{X}^{PBE,LR}\left(\omega \right)$ is corresponding to a long-range component of PBE and $E_{C}^{PBE}$ related to PBE correlated functional, respectively.

To model all of the proposed systems, the periodic boundary conditions (PBC) were used along all axes; in this way, it is possible to avoid problems with boundary effects caused by the finite size and to reduce the calculation time while maintaining high accuracy. The energy cut-off has been fixed at 1200 eV and the Brillouin-zone integration has been performed over a 15 × 15 × 15 k-points grid for the modelled *P2_1_/c*, *Pca2_1_*, and *Fm*$\bar{3}$*m* polymorphs. These parameters assure the total energy convergence of 5.0 × 10^−6^ eV/atom, the maximum stress of 2.0 × 10^−2^ GPa, and the maximum displacement of 5.0 × 10^−4^ Å. The modern theory of polarization [37] and the Berry phase operator method were used to obtain the polarization in the respective polymorphs. The total polarization is the sum of the electronic (*P_e_*) and ionic (*P_i_*) contributions.

The electronic one (*P_e_*) has been calculated as in Equation (2):(2)$$P_{e}=-\frac{2\left|e\right|i}{{\left(2\pi \right)}^{3}}\int_{A}dk_{⊥}\sum_{n=1}^{M}\int_{0^{\langle U_{k,n}\left|\frac{\partial }{\partial k}\right|u,<,n\rangle }}^{G}dk$$
where the sum runs over occupied bands and *k* and the direction of polarization are parallel to each other. The *G* term is a reciprocal lattice vector in the same direction. The states *U_k,n_* > are the cell-periodic parts of the Bloch functions *y_k,n_* (*r*) = *u_k,n_* (*r*) *e^ikr^*. The last integral is known as the Berry phase [38].

The ionic contribution (*P_i_*) has been calculated using a simple classical electrostatic sum of point charges, as reported in Equation (3):(3)$$P_{i}=\frac{\left|e\right|}{\Omega }\sum_{\nu }Z_{ion}^{v}r^{v}$$
where *Ω* is the unit cell volume, *Z^v^_ion_* is the valence charge, and *r^ν^* is the position vector of the *ν* atom.

To evaluate the geometrical stability of the different polymorphs, we calculated the cohesive energy per atom, using Q-ATK software, using the following Equation (4):(4)$$E_{cohesive}=\frac{E_{total}-\left(n_{Hf}E_{Hf}+n_{O}E_{O}+n_{Y}E_{Y}\right)}{M}$$
where *E_total_* is the total energy of each structure, *E_Hf_*, *E_O_*, and *E_Y_* are the total energy of the single isolated atoms (Hf, Y, or O) in the same crystalline structure, *n_Hf_*, *n_O_*, and *n_Y_* are the total number of atoms (Hf, Y, or O), and *M* is the total number of atoms in the unit cell.

The optical properties of the HfYO_2_ structures were determined by two components of the dielectric function *ε*(*ω*) = *ε*_1_(*ω*) + *iε*_2_(*ω*).

The imaginary part *ε*_2_ (*ω*) of dielectric constant can determine from Equation (5) [39,40,41]:(5)$$\varepsilon_{2}\left(\omega \right)=\frac{4\pi^{2}}{\Omega \omega^{2}}\underset{i\in \mathrm{HOMO},j\in \mathrm{LUMO}}{\sum }\underset{k}{\sum }W_{k}{\left|\rho_{ij}\right|}^{2}\delta \left(\varepsilon_{kj}-\varepsilon_{ki}-\hbar \omega \right)$$
where HOMO, LUMO, *ω*, Ω, *W_k_*, *ρ_ij_* were the valence band, conduction band, photon frequency, volume of the lattice, weight of the *k*-point, and elements of the dipole transition matrix, respectively.

The real part of the dielectric constant can be obtained with following Equation (6):(6)$$\varepsilon_{1}\left(\omega \right)=1+\frac{1}{\pi }P\int_{0}^{\infty }d\bar{\omega }\frac{\bar{\omega }\varepsilon_{2}\left(\bar{\omega }\right)}{{\bar{\omega }}^{2}-\omega^{2}}$$

Finally, the refractive index of HfYO_2_ structures has been calculated as follows (7):(7)$$R\left(\omega \right)={\left|\frac{\sqrt{\varepsilon \left(\omega \right)+1}}{\sqrt{\sqrt{\varepsilon \left(\omega \right)-1}}}\right|}^{2}$$

## 3. Results and Discussion

### 3.1. Geometrical and Lattice Parameters

The crystalline structure of HfO_2_ is characterized by precise geometrical parameters based on the nature of polymorphs; in fact, depending on the synthetic procedure [42,43], different phases can be induced during HfO_2_ formation: the monoclinic (m-) with a space group *P2_1_/c*, the orthorhombic (o-) with a space group *Pca2_1_*, and the cubic phase (c-) with a space group *Fm*$\bar{3}$*m* are some examples. In order to understand the effect of Yttrium on the geometrical parameters, the three polymorphs were used as starting geometry and further optimized before (pure HfO_2_) and after the addition of 8% of Y, 12% of Y, and 16% of Y; all the systems under study are reported schematically in Figure 1. Y atoms were added to the systems before the geometry optimization and by replacing a few Hf atoms according to the doping percentage. The number of atoms and the dimension of the supercell of the monoclinic systems were: (i) m-HfO_2_ was composed of 12 atoms, the vectors were (a) 5.116 Å, (b) 5.172 Å, and (c) 5.295 Å; (ii) m-HfO_2_ with 8% of Y was composed of 36 atoms, the vectors were (a) 5.116 Å, (b) 5.172 Å, and (c) 15.885 Å. (iii) m-HfO_2_ with 12% of Y was composed of 24 atoms, the vectors were (a) 5.116 Å, (b) 5.172 Å, and (c) 10.590 Å (iv) m-HfO_2_ with 16% of Y was composed of 36 atoms, the vectors were (a) 5.116 Å, (b) 5.172 Å, and (c) 15.681 Å. The dimensions of the supercell of the cubic systems were (a) = (b) = (c) 5.115 Å in HfO_2_; in both 8% and 16% Y:HfO_2_ the super-cell dimension were (a) = (b) 5.115 Å and (c) 15.345 Å; while in the cubic system with 12% Y:HfO_2_ the dimensions were (a) = (b) 5.115 and (c) 10.230 Å. The total number of atoms in the cubic systems was 3, 36, 24, and 36, respectively, for HfO_2_ with 0%, 8%, 12%, and 16% of Y doping. The number of atoms and the dimension of the supercell of the orthorhombic systems were: (i) o-HfO_2_ was composed of 12 atoms, the vectors were (a) 5.231 Å, (b) 5.008 Å, and (c) 5.052 Å; (ii) o-HfO_2_ with 8% of Y was composed of 36 atoms, the vectors were (a) 5.243 Å, (b) 5.063 Å, and (c) 15.237 Å; (iii) o-HfO_2_ with 12% of Y was composed of 24 atoms, the vectors were (a) 5.243 Å, (b) 5.063 Å, and (c) 10.158 Å; (ii) o-HfO_2_ with 16% of Y was composed of 36 atoms, the vectors were (a) 5.243 Å, (b) 5.063 Å, and (c) 15.237 Å.

In the same way, Table 1 reports the values of the calculated lattice parameters for the monoclinic, the orthorhombic, and the cubic configurations of (i) HfO_2_ with 0% of Y substitution, (ii) HfO_2_ with 8% of Y substitution, (iii) HfO_2_ with 12% of Y substitution, and (iv) HfO_2_ with 16% of Y substitution to the total amount of Hf elements.

Lattice energies minimized for the m-HfO_2_ *P2_1_/c* polymorph were obtained by optimization of the atomic positions and altering the size and angle of the unit cell, systematically. After optimization of the lattices, unit cell dimensions (Table 1) of 5.116 Å, 5.172 Å, and 5.295 Å are found for a, b, and c, respectively, in line with what has already been reported in the literature [17,44]. The lattice vectors remain the same when moving from the undoped HfO_2_ to the 8% and 12% of Y doping, while the c vector slightly decreases by imposing the 16% of Y in the system. The lengths of the bonds between Hf-O were 2.16 Å, 2.19 Å, 2.21 Å, and 2.21 Å when moving from 0 to 16% of doping; similarly, the lengths of the Y-O bonds were 2.23 Å, 2.30 Å, and 2.29 Å, respectively, for 8%, 12% and 16% of Y doping. The angles between O-Hf-O were 103° for all the considered systems, while the angles O-Y-O were 97°, 95°, and 93° for 8%, 12%, and 16% of doping percentages. From these data, Y has the property to reduce the angle bonds with O, which is due to the M^+3^ nature of Y as a dopant, since it has 1 d electron less then Hf.

The orthorhombic *Pca2_1_* polymorph is directly related to the ferroelectricity behavior of HfO_2_, which is due to the formation of a non-centrosymmetric polar phase. The optimized lattices for o-HfO_2_ were 5.231 Å, 5.008 Å, and 5.052 Å for a, b, and c vectors, respectively (Table 1), which is in line with previous results [11,17,24,45].

The effect of Y on the *Pca2_1_* unit cell is generally more evident than that observed for *P2_1_/c* polymorphs. In particular, all the doped systems show a sensitive increase of b and c vectors [45]. More, the amount of Y incorporated in the unit cell it is not related to the change, in other words, the doping of HfO_2_ with Y, from 8% to 16%, affects the unit cell lattice vector in the same way. The length of the bond between Hf-O were 2.14 Å, 2.15 Å, 2.16 Å, and 2.16 Å when moving from 0 to 16% of doping; similarly, the length of the Y-O bonds was 2.23 Å, 2.18 Å, and 2.22 Å, respectively, for 8%, 12% and 16% of Y doping. The angles between O-Hf-O were 106° in HfO_2_, 104° in 8% Y:HfO_2_, 101° in 12% Y:HfO_2_, and 102° in 16% Y:HfO_2_; while the angles O-Y-O were more affected by the doping percentage, in fact, they moved from 83°, 80°, to 78° for 8%, 12%, and 16% of Y content. Also in this case, the Y presence led to a decrease of the angle-bonds, more, since the *Pca21* polymorph is an asymmetric phase, this effect seems to be more evident respect to the angle-bonds variations detected for the m-polymorph.

The c-HfO_2_ *Fm*$\bar{3}$*m* polymorph shows the same lattice vector independently of the presence of Y; the optimized lattices for the systems are 5.115 Å, 5.115 Å, and 5.115 Å for a, b, and c vectors, respectively. For this polymorph, the length of the bond between Hf-O was 2.21 Å, 2.12 Å, 2.23 Å, and 2.19 Å when moving from 0 to 16% of Y amount in HfO_2_; while the length of the Y-O bonds was 2.34 Å, 2.31 Å, and 2.33 Å, respectively, for 8%, 12% and 16% of Y doping. The angles between O-Hf-O were 109° in HfO_2_, 110° in 8% Y:HfO_2_, 111° in 12% Y:HfO_2_, and 111° in 16% Y:HfO_2_. The angles in between O-Y-O atoms move from 109°, 112°, to 114° for 8%, 12%, and 16% of Y content showing a modification as the percentage of Y increases. In this case, an opposite trend on the angle bonds were observed when Y is present, since a gradual increase can be observed gradually adding Y. This is due to a more evident accommodation of the atoms in this phase since the lattice vectors never changed.

### 3.2. Ground State Cohesive Energy

In order to get a deeper insight into the behavior of the phases and to underline the role of Y as a dopant, the PBE/GGA method is used to calculate the ground state cohesive energy of all the examined polymorphs (Figure 2) using Q-ATK code. The cohesive energy is the difference between the energy per atom of the bulk material at equilibrium and the energy of a free atom in its ground state; a more negative energy indicates a more stable structure.

It is well known that the monoclinic structure of HfO_2_ is a stable polymorphic phase for this material [46], and the calculated cohesive energy for m-HfO_2_ confirms this experimental evidence. The direct comparison between pure m-HfO_2_ and the doped monoclinic systems indicates a decrease of stability; in fact, the energy values move from −8.97 in HfO_2_ to −8.80, −8.72, and −8.13 in HfO_2_ with 8%, 12% and 16% of Y, respectively. Thus, the energy of the monoclinic phase is slightly destabilized by the presence of Y as a doping agent. A similar trend is observed for the orthorhombic polymorph, even if the starting cohesive energy of the pure HfO_2_ is less negative (i.e., the orthorhombic system is more unstable if compared to the respective monoclinic); as said, the presence of Y acts exactly in the same way: moving from o-HfO_2_ without Y to the one with 8%, 12%, and 16% of Y, the energies change from −8.80 to −8.72, −8.68 and −8.19, respectively. Instead, the cubic polymorph has a behavior that differs from the previously described monoclinic and orthorhombic phases; in the cubic structures the formation energy for the undoped HfO_2_ has the same value found for the o-HfO_2_, both are less negative than that of the m- structure, confirming that monoclinic remains the most stable between the analyzed phases. However, after the addition of 8% of Y to the structure, the energies decrease (i.e., increase the stability). The values obtained are −8.80 and −8.86 for HfO_2_ and 8% of Y in HfO_2_, respectively. The systems with a higher Y% are again destabilized: the cohesive energy associated with a doping of 12% and 16% are −8.63 and −8.16, respectively. The cubic phase of these last two systems containing Y has comparable cohesive energy to the respective monoclinic and orthorhombic polymorphs. The cohesive energy calculated indicates how the monoclinic polymorph remains the preferential phase for pure HfO_2_, while the presence of Y destabilizes the structures in any polymorphs, and in any doping percentage, except for the cubic HfO_2_ with 8% of Y. This is not surprising, because, as already reported by other studies, different polymorphs can be stabilized by doping with Al [47], Si [19], Zr [17], or Y [23]. It has been proven that the presence of Y inside the HfO_2_ created oxygen vacancies, and consequently, the energy of the cubic phases is reduced [20,48]. As reported by other studies, the cubic HfO_2_ was fully stabilized at a minimum concentration of 8.7 mol% of YO_1.5_ [23]. Accordingly, Chen et al. [21] attest that the concentration of Y_2_O_3_ affects the crystallization of HfO_2_-doped film; the cubic phase of the film appears at a doping ratio of 8 mol% without a post-annealing procedure. Amorphous and monoclinic phases of HfO_2_ are stable at room temperature, while the transformation to cubic or orthorhombic polymorphs (higher-k phases), typically arises at higher temperatures (e.g., 2900 K for cubic phase) [19], which are poorly compatible with the common manufacturing procedures. However, stabilization of higher-k dielectric HfO_2_ at lower temperatures could be helpful in electronic applications. The stabilization of the high-temperature crystalline structure at lower temperatures and ambient conditions can be realized by doping [21,22]; The formation cohesive energy calculated for all the presented systems demonstrates the stabilization of the cubic polymorphs by adding 8% of Y in HfO_2_. Finally, the calculations were repeated using QE code, and the same results were obtained. This confirmed the reliability of the obtained results.

### 3.3. Dielectric Constant and Optical Band Gap

The effects of Y on the real part (ε_r_) of the dielectric function, which describes the ability of the matter to interact with an electric field without absorbing energy, for the m- *P2_1_/c*, the o- *Pca2_1_*, and the c- *Fm*$\bar{3}$*m* polymorphs were calculated on a wide-energy range and reported in Figure 3. The values were reported in function of a simulated electromagnetic field applied on materials, and the energy associated was expressed in eV.

As shown, the ε_r_ values calculated for HfO_2_ are 5.42, 5.75, and 5.93 for the monoclinic, orthorhombic, and cubic phase, respectively; the results are in line with what has been previously reported [17]. The presence of Y in HfO_2_ at 8% brings an increase of the dielectric constant values; in particular, the ε_r_ of the monoclinic polymorph reaches 28.27*,* the orthorhombic one extends to 35.54, and similarly, the cubic phase reaches 36.81. This is not surprising, since it is already known that Y promotes the transition to a higher dielectric constant [21,22,23,27].

All these values are in line with the results presented by Liang et al. [23], which reported the variation of the relative dielectric constant and cubic phase fraction as a function of Y content; in particular, the cubic HfO_2_ was fully stabilized at a minimum concentration of 8.7 YO_1.5_ with a relative dielectric constant value of 32.

Similarly, the dielectric constant reaches a value of 32 as the Y concentration is 8.7 mol% for a film of about 10 nm of Hf-Y-O [22].

As the % of Y increases, the dielectric constant shows a different trend; in particular, for the system containing 12% of Y in HfO_2_ the dielectric constants of monoclinic, orthorhombic, and cubic phases are 41.39, 51.51, and 84.97, respectively. The HfO_2_ doped with 16% of Y has ε_r_ at 0 eV of 64.76, 65.23, and 175.92 for monoclinic, orthorhombic, and cubic crystalline structures. The higher percentage of Y seems to destabilize the structure of the HfO_2_ in the cubic phase, and the dielectric constant value is overestimated.

As reported by other studies [22,23], a doping concentration of about 8% of Y brings a stabilization of the cubic phase in HfO_2_. In our case, moving to a higher concentration of Y-dopant than the minimum value required to stabilize the cubic polymorph affects the geometry and the energy of the structures, and as a consequence, the results obtained for the 12% and 16% of Y doping are not so accurate as those obtained for the 8% of Y.

The calculated imaginary part of the dielectric function (Figure 4), which describes the ability of the matter to permanently absorb energy from a time-varying electric field, predicts an absorption peak in the ultraviolet region associated with a static dielectric constant of 5.7, 5.8, and 5.5 eV for the monoclinic, orthorhombic, and cubic HfO_2_, respectively; these values represent the estimation of the optical bandgap energy. To report the same unit measure, in this case, the values were also reported in function of a simulated electromagnetic field applied on materials, expressed in eV. It is important to underline that our approach predicts bandgaps which are perfectly in line with the experimental value of 5.7 eV proposed by Balog et al. [49] moreover, our values better approximate the theoretical data proposed by Koller et al. [13,50] and Jaffe et al. [13], which underestimated the band gap of m- and c-HfO_2_ by using PBE and GGA approximation, respectively.

Our results are not dissimilar to what was reported by Padilha and McKenna [27], even if their values are slightly overestimated (band gap of 6.4 and 6.2 eV, for m- and c- HfO_2_, respectively).

Similarly, the band gap calculated by HSE hybris functional for the systems containing the 8% of Y brings to an increase of the band gap to 5.8, 6.3, and 6.0 eV for the m-, o-, and c- polymorphs, respectively. The increasing of Y inside the HfO_2_ structures does not systematically or drastically affect the estimated optical band gap, even if the band gap results are higher when 16% of Y replaces Hf in HfO_2_. In detail, the systems containing 12% of Y display a band gap for monoclinic, orthorhombic, and cubic phases of 6.0, 6.1, and 5.8 eV, respectively. The HfO_2_ doped with 16% of Y presents a band gap of about 6.0, 6.8, and 6.5 eV correspondingly to monoclinic, orthorhombic, and cubic crystalline structures.

### 3.4. Refractive Index

The effects of Y on HfO_2_ have also been used to understand the trend of the refractive index, which is useful to understand the ability of the matter to bent or refract the light that enters inside the material itself. Also in this case, the values were reported in function of a simulated electromagnetic field applied on materials expressed in eV. Again, all three polymorphs: monoclinic *P2_1_/c*, orthorhombic *Pca2_1_*, and cubic *Fm*$\bar{3}$*m* polymorphs were considered and the refractive index (*n*), calculated on a wide-energy range, is reported in Figure 5.

The systems based on pure and 8% Y-doped HfO_2_ show a refractive index that only slightly depends on the polymorphs considered, in particular, in the low energy range; on the contrary, the 12% and 16% Y:HfO_2_ structures display a refractive index dependent on the considered phase, at low energy. The cubic phase has higher values of *n* below 0.5 eV.

In order to compare the data to what was already reported by Chen et al. [21], the value of the refractive index at 3 eV (about 400 nm) is plotted as a function of the different percentages of Y in HfO_2_, as well as the polymorphs. The systems show only minor differences in terms of *n* at 3 eV, and the values are in good agreement with those already reported. Moreover, a decrease of the refractive index is detected when moving from the system without the Y and the Y-doped ones.

## 4. Conclusions

In this paper, we exploited three different crystalline structures of HfO_2_: the monoclinic *P21/c*, the orthorhombic *Pca2_1_*, and the cubic *Fm*$\bar{3}$*m* structures. Each polymorph is characterized by singular properties that can also be altered by doping elements in the unit cell. We reported the study of twelve different systems characterized by the three diverse polymorphs reported and doping percentages. For each polymorph, 0% Y:HfO_2_, 8% Y:HfO_2_, 12% Y:HfO_2_, and 16% Y:HfO_2_ were modeled and studied. The density functional theory (DFT) method based on PBE-GGA was used to optimize the geometries, calculate the real part of the dielectric constant, and estimate the refractive index. Moreover, the HSE hybrid functionals were used to predict the imaginary part of the dielectric constant, and thus, the optical bandgap energy. Results showed that Y affects the formation energy in different ways and causes changes in the optical properties depending on the polymorphs. When the percentage of Y did not exceed 12%, a stabilization of the cubic phase fraction and an increase of the dielectric constant were observed. The calculated optical results obtained by HSE indicated a very good agreement with the experiments. While the real part of the dielectric constant of different polymorphs with 8% Y showed values of 36.81, 35.54, and 28.27 predicted for the cubic, the orthorhombic, and the monoclinic structures, the imaginary part of the dielectric constant revealed perfect optical absorption in the infrared and ultraviolet regions of the electromagnetic light. Moreover, the energy band gap values are in perfect agreement to what was already reported by other theoretical papers; however, our calculations best matched the experimental findings. Only minor differences are found between the three polymorphs in terms of both refractive index and optical band gap. The adopted first principles study verifies the available experimental data, identifies the effects of doping phenomena, and generates a reasonable prediction of the physical-chemical properties of all the systems, allowing for control of the properties of the materials at nanoscale.

## Figures and Tables

**Figure 1 nanomaterials-12-04324-f001:**
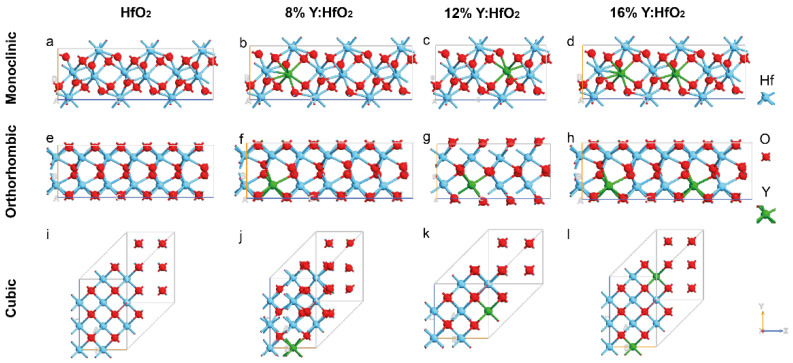
Schematic representation of all the examined systems. Monoclinic *P21/c* (**a**) HfO_2_, (**b**) 8% Y:HfO_2_, (**c**) 12% Y:HfO_2_, and (**d**) 16% Y:HfO_2_; Orthorhombic *Pca21* (**e**) HfO_2_, (**f**) 8% Y:HfO_2_, (**g**) 12% Y:HfO_2_, and (**h**) 16% Y:HfO_2_; Cubic *Fm$\bar{3}$m* (**i**) HfO_2_, (**j**) 8% Y:HfO_2_, (**k**) 12% Y:HfO_2_, and (**l**) 16% Y:HfO_2_. Color code in the ball and stick model: Y green, Hf blue and O red.

**Figure 2 nanomaterials-12-04324-f002:**
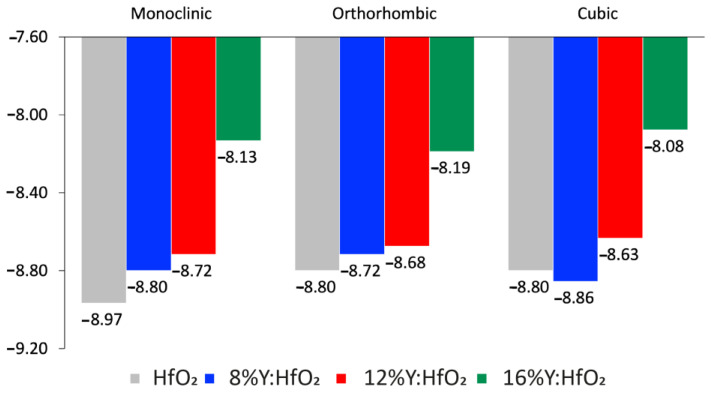
Ground state cohesive energy formation. Monoclinic *P2_1_/c,* orthorhombic *Pca2_1_*, and cubic *Fm$\bar{3}$m* polymorphs of HfO_2_ (grey bar), 8% Y:HfO_2_ (blue bar), 12% Y:HfO_2_ (red bar), and 16% Y:HfO_2_ (green bar).

**Figure 3 nanomaterials-12-04324-f003:**
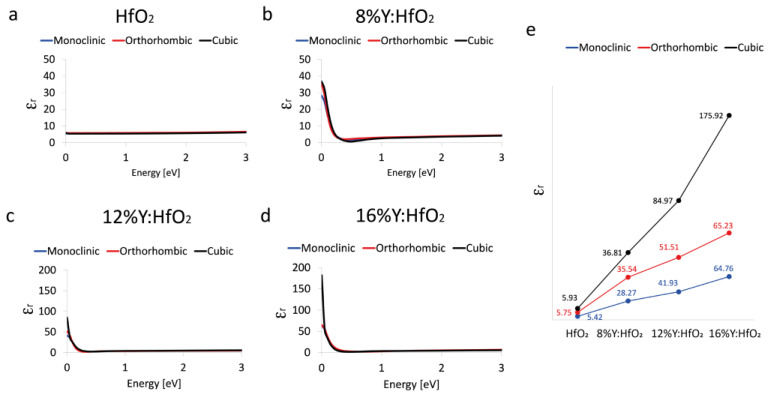
Real part of the dielectric constant. ε_r_ of monoclinic *P2_1_/c* (blue curve), orthorhombic *Pca2_1_* (red curve), and cubic *Fm$\bar{3}$m* (black curve) polymorphs evaluated between 0 and 3 eV of HfO_2_ (**a**), 8% Y:HfO_2_ (**b**), 12% Y:HfO_2_ (**c**), and 16% Y:HfO_2_ (**d**). The schematic trend of ε_r_ at 0 eV as a function of the crystalline phase and Y content (**e**).

**Figure 4 nanomaterials-12-04324-f004:**
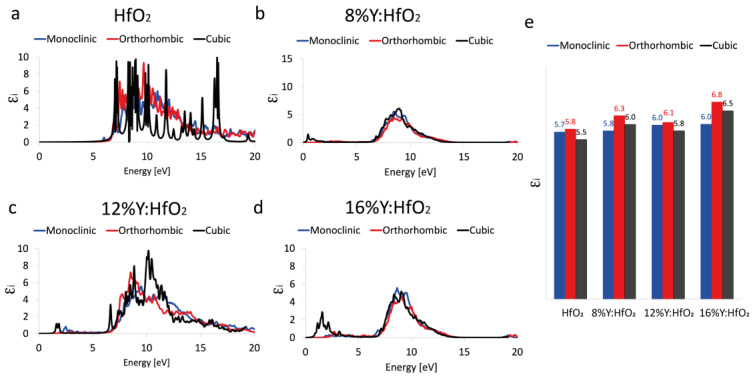
Imaginary part of the dielectric constant. ε_i_ of monoclinic *P2_1_/c* (blue curve), orthorhombic *Pca2_1_* (red curve), and cubic *Fm$\bar{3}$m* (black curve) polymorphs evaluated between 0 and 20 eV of HfO_2_ (**a**), 8% Y:HfO_2_ (**b**), 12% Y:HfO_2_ (**c**), and 16% Y:HfO_2_ (**d**). Schematic trend of ε_i_ at the main absorption peak as a function of the crystalline phase and Y content (**e**).

**Figure 5 nanomaterials-12-04324-f005:**
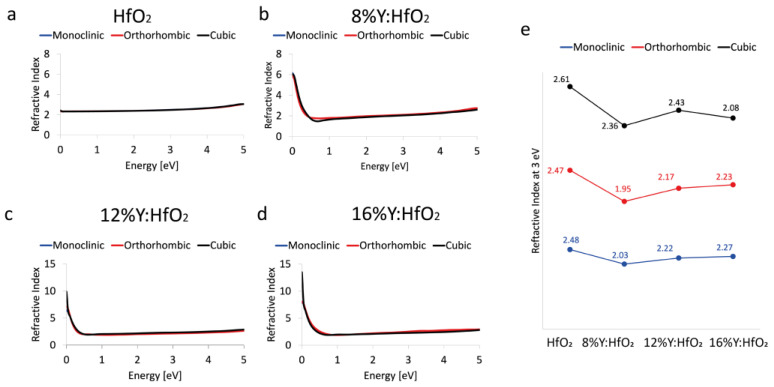
Refractive Index. Refractive index of monoclinic *P2_1_/c* (blue curve), orthorhombic *Pca2_1_* (red curve), and cubic *Fm$\bar{3}$m* (black curve) polymorphs evaluated between 0 and 5 eV of HfO_2_ (**a**), 8% Y:HfO_2_ (**b**), 12% Y:HfO_2_ (**c**), and 16% Y:HfO_2_ (**d**). Schematic trend of *n* at 3 eV as a function of the crystalline phase and Y content (**e**).

**Table 1 nanomaterials-12-04324-t001:** Lattice vectors for monoclinic *P2_1_/c,* orthorhombic *Pca2_1_*, and cubic *Fm$\bar{3}$m* polymorphs of HfO_2_, 8% Y:HfO_2_, 12% Y:HfO_2_, and 16% Y:HfO_2_.

	Å	HfO_2_	8% Y:HfO_2_	12% Y:HfO_2_	16% Y:HfO_2_
Monoclinic *P2_1_/c*	a	5.116	5.116	5.116	5.116
	b	5.172	5.172	5.172	5.172
	c	5.295	5.295	5.295	5.227
Orthorhombic *Pca2_1_*	a	5.231	5.243	5.243	5.243
	b	5.008	5.063	5.063	5.063
	c	5.052	5.079	5.079	5.079
Cubic *Fm*$\bar{3}$*m*	a	5.115	5.115	5.115	5.115
	b	5.115	5.115	5.115	5.115
	c	5.115	5.115	5.115	5.115

## Data Availability

Not applicable.

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
