# Peer review of "The Effect of Y Doping on Monoclinic, Orthorhombic, and Cubic Polymorphs of HfO_2_: A First Principles Study"

_nanomaterials, 2022, doi:10.3390/nano12234324_

Round 1

Reviewer 1 Report

In this contribution the authors reported in-depth calculations of Y doping effect on the cohesive energy and optical properties of various polymorphs of HfO2.  This study was well carried out and scientifically sound and thus suitable for publication in Nanomaterials. There are some minor issues that need to be addressed:

1.     In the Introduction the authors mentioned phase transition as a motivation but did not fully investigate.  Is it possible to investigate a few of them which have significance in some applications?  If it requires too many calculations, could the authors provide some relevant references?

2.     Section 3.2 needs to have the equation for cohesive energy; or simply to add a sentence e.g. “more negative energy indicates more stable…”

3.     Any G(T,p) calculations of Gibbs free energy?  It might have some implications for phase transition and can be added to the discussion.   

4.     “The schematic trend 246 of er at 0 eV as a function of the crystalline phase and Y content (e)” – why at 0 eV?  The reviewer is not an expert in optical properties; It would be great to explain what are the x-axis in Figure 3,4,5 before going into discussion.  

5.     In the manuscript there were mixed use of “first principle” and “first principles”.  The reviewer believes the latter should be used.

6.     Minor comment: in Figure 1, the color of Y is too similar to that of Hf.  Perhaps it could be changed to some other color.

Author Response

In this contribution the authors reported in-depth calculations of Y doping effect on the cohesive energy and optical properties of various polymorphs of HfO2.  This study was well carried out and scientifically sound and thus suitable for publication in Nanomaterials. There are some minor issues that need to be addressed:

Author: We would like to thank the Referee for the comments and ideas suggested for improving our manuscript. In the following line, we kindly answer your questions and observations. The changes have been made to the text are in red.

  1. In the Introduction the authors mentioned phase transition as a motivation but did not fully investigate.  Is it possible to investigate a few of them which have significance in some applications?  If it requires too many calculations, could the authors provide some relevant references? Author: Thank you for your comment. HfO2 and its doped derivatives are characterized by a strong relationship between some important properties (i.e. optical or electronic) and the polymorph type. Phase transition cannot be investigated by DFT calculation, since we should include a term to directly simulate the effects of temperature. For this reason, to simulate the phase transition, ab initio molecular dynamics should be used, but the calculations are different from those reported. To better explain the phase transition, some references have been added to the Introduction section and a few examples of applications have been reported.

  1. Section 3.2 needs to have the equation for cohesive energy; or simply to add a sentence e.g. “more negative energy indicates more stable…” Author: thank you for your valuable comment, we added the suggested sentence in Section 3.2 and also the equation (4) for cohesive energy in the Methods.

  1. Any G(T,p) calculations of Gibbs free energy?  It might have some implications for phase transition and can be added to the discussion. Author: thank you very much for this suggestion. This is of course an interesting point to consider. In general, the software that we have used cannot directly give the Gibbs free energy (G), since the Gibbs free energy of materials is strictly correlated to the pressure P and the Helmholtz free energy F, which is a function of temperature and volume. Formally, for the Gibbs free energy, a volume term PV should be included, where P is the external pressure, and V is the system volume. This might make sense if we want to compare the free energies at different volumes, although the PV correction is typically tiny, but this is not our case, since we always have bulk systems with fully periodic boundary conditions, and not semi-infinite crystals. For all these reasons, we cannot provide calculations of Gibbs free energy for these systems using this software, but, in any case, this investigation would not give us more information than we get with the cohesive energy calculation for these systems.

  1. “The schematic trend 246 of er at 0 eV as a function of the crystalline phase and Y content (e)” – why at 0 eV?  The reviewer is not an expert in optical properties; It would be great to explain what are the x-axis in Figure 3,4,5 before going into discussion. Author: thank you for your comment, which gives us the possibility to better describe the reported figures. We decided to compare the trend of the real part of the dielectric constant at 0 eV since in that part the major changes related to different polymorphs and Y content are present.

Figure 3, Figure 4, and Figure 5 report the real part of the dielectric constant (er), the imaginary part of the dielectric constant (ei), and the refractive index (n), respectively. All these three physical parameters described are dimensionless. Anyway, a description about the x-axis of these figures has been added in the manuscript as the reviewer suggested. In general, the calculated values have been reported in function of a simulated electromagnetic field applied on materials, expressed in eV.

  1. In the manuscript there were mixed use of “first principle” and “first principles”.  The reviewer believes the latter should be used. Author: thank you for the comment. We uniformed “first principles” in the manuscript.

  1. Minor comment: in Figure 1, the color of Y is too similar to that of Hf.  Perhaps it could be changed to some other color. Author: thank you again, the reviewer is right. We changed the color of Y in Figure 1.

Reviewer 2 Report

The authors very clearly describe the purpose of their article. The introduction provides sufficient information about the field of activity. The results obtained are interesting, but their description and interpretation could be improved. As well, the methods that were used for this research have to be described more explicitly. English editing is also required. Paper cannot be accepted in present form and requires a major revision.

1) Figure 1. Please add more contrast to Y atoms, blue and light blue is difficult to distinguish.

2) Please, describe the models in a bit more details. What is the size of the supercell, how many atoms are there. When dopants were added, how the symmetry of the unit cell was changed? I am curious, why such high cutoff energy was selected, and also it looks like there are so many k-points for large supercells. Or were that parameters only for 0%Y bulk?

3) You do not mention, are your calculations closed-shell or open-shell? In case of Y doping, which will be Y3+, there should be a hole localized on the nearest oxygen atoms and that might affect the results. This paper https://doi.org/10.1021/acs.chemmater.5b00800 shows the effect on the structure of m-HfO2 of M3+ dopants, so probably this should be taken into account.

4) For calculation of cohesive energy you need energies of single atoms. In Methods you mention 3 different codes: Q-ATK, QE and SIESTA. Can you explicitly tell which code was used for what calculations? In case you compare energies from different codes. this also should be mentioned.

5) You do not mention how you did calculate the refractive index. That should be added in Methods section.

Author Response

The authors very clearly describe the purpose of their article. The introduction provides sufficient information about the field of activity. The results obtained are interesting, but their description and interpretation could be improved. As well, the methods that were used for this research have to be described more explicitly. English editing is also required. Paper cannot be accepted in present form and requires a major revision.

Author: We would like to thank the Referee for the comments to improve our manuscript. In the following line, we kindly answer your questions and observations. The changes made to the text are in red.

1) Figure 1. Please add more contrast to Y atoms, blue and light blue is difficult to distinguish.

Author: thank you for your suggestion, we changed the color of Y in Figure 1.

2) Please, describe the models in a bit more details. What is the size of the supercell, how many atoms are there. When dopants were added, how the symmetry of the unit cell was changed? I am curious, why such high cutoff energy was selected, and also it looks like there are so many k-points for large supercells. Or were that parameters only for 0%Y bulk?

Author: thank you very much, indeed, we did not describe in detail the modeling and the set up. The cutoff energy and k-points used are the same for all the studied systems. The choice of the high cutoff energy is due to the use of Q-ATK and QE codes, since we must consider two different implementations. These functions are discretized on numerical grids, and if we high frequency (small wavelength) planewaves, they will therefore require increasingly finer grid points to be described. Of course, this approach increased computational cost, but in this way, we can be sure that we had reliable results using both the codes. In a similar way we reasoned for the k-points. About the size of the supercell, the number of atoms, and when the dopants have been added, we described all these parts more in detail in the 3.1 section.

3) You do not mention, are your calculations closed-shell or open-shell? In case of Y doping, which will be Y3+, there should be a hole localized on the nearest oxygen atoms and that might affect the results. This paper https://doi.org/10.1021/acs.chemmater.5b00800 shows the effect on the structure of m-HfO2 of M3+ dopants, so probably this should be taken into account.

Author: thank you for your interesting suggestion. This is a very important step to consider. In the manuscript, we did not distinguish between closed-shell and open-shell because we also described the pure HfO2, in which we have an even number of valence electrons, considering Hf+4 and O-2 entities. Of course, the incompletely 1d electron less subshell of Y makes its interactions different with respect to that observed for Hf. Indeed, in our simulations we always describe the overlap of wavefunctions that generates chemical bonds, then, the local phenomena around Y have been considered and the differences in cohesive energy reported are strictly correlated to Y presence. The Y shows an important impact in the changes of bond angles in HfO2 polymorphs. In general, we observed a gradual decrease of angle bonds for monoclinic and orthorhombic phases when we gradually increased the Y percentage. The opposite trend has been observed for the cubic phase. These phenomena are correlated to the M+3 nature of Y as dopant. We added detailed descriptions of these effects in 3.1 section.

4) For calculation of cohesive energy you need energies of single atoms. In Methods you mention 3 different codes: Q-ATK, QE and SIESTA. Can you explicitly tell which code was used for what calculations? In case you compare energies from different codes. this also should be mentioned.

Author: thank you very much, with this suggestion we can better clarify what methods we have used. In detail, we never used SIESTA, we reported it in the manuscript because it is well known in literature and its formalism is like those we used, then, this is just a comparison. We calculated the cohesive energy values using Q-ATK code, but we also found that the results were the same when we used QE. We better report this part in the 3.2 Section. More, we added equation (4) for cohesive energy calculation in the Methods to better clarify our approach.

5) You do not mention how you did calculate the refractive index. That should be added in Methods section.

Author: thank you very much for your valuable comment. We added all the required formulas for the calculations. Since the refractive index is strictly correlated to the real and imaginary parts of the dielectric constant, we reported all the equations that allowed us to determine the refractive index.

Round 2

Reviewer 2 Report

All issues have been addressed. The paper is now suitable for publication in Nanomaterials.